# Measurement and Simulation of Ultra-Low-Energy Ion–Solid Interaction Dynamics

**DOI:** 10.3390/mi14101884

**Published:** 2023-09-30

**Authors:** Michael Titze, Jonathan D. Poplawsky, Silvan Kretschmer, Arkady V. Krasheninnikov, Barney L. Doyle, Edward S. Bielejec, Gerhard Hobler, Alex Belianinov

**Affiliations:** 1Ion Beam Laboratory, Sandia National Laboratories, Albuquerque, NM 87185, USA; 2Center for Nanophase Materials Sciences, Oak Ridge National Laboratory, Oak Ridge, TN 37831, USA; 3Institute of Ion Beam Physics and Materials Research, Helmholtz-Zentrum Dresden-Rossendorf, 01328 Dresden, Germany; 4Institute of Solid-State Electronics, TU Wien, Gußhausstraße 25-25a, A-1040 Wien, Austria

**Keywords:** focused ion beam, ion implantation, ultra-low energy

## Abstract

Ion implantation is a key capability for the semiconductor industry. As devices shrink, novel materials enter the manufacturing line, and quantum technologies transition to being more mainstream. Traditional implantation methods fall short in terms of energy, ion species, and positional precision. Here, we demonstrate 1 keV focused ion beam Au implantation into Si and validate the results via atom probe tomography. We show the Au implant depth at 1 keV is 0.8 nm and that identical results for low-energy ion implants can be achieved by either lowering the column voltage or decelerating ions using bias while maintaining a sub-micron beam focus. We compare our experimental results to static calculations using SRIM and dynamic calculations using binary collision approximation codes TRIDYN and IMSIL. A large discrepancy between the static and dynamic simulation is found, which is due to lattice enrichment with high-stopping-power Au and surface sputtering. Additionally, we demonstrate how model details are particularly important to the simulation of these low-energy heavy-ion implantations. Finally, we discuss how our results pave a way towards much lower implantation energies while maintaining high spatial resolution.

## 1. Introduction

Ion implantation has been a workhorse for the semiconductor industry [1,2,3,4]. However, with technological scaling towards smaller devices and shallower junctions, implant energies must decrease [5,6,7,8]. Additionally, as novel materials are introduced to make next-generation devices, including quantum information science (QIS) devices, a larger ion selection becomes necessary [9,10,11,12,13,14,15]. Finally, with the projected integration of 2D materials into future semiconductors, the implants must be made within a single atomic layer and at a precise location [16,17]. To make 2D semiconductor-based devices, and to enable higher throughput quantum devices, quantitative, spatially precise, and deterministic placements of new ion species—not yet explored by the semiconductor industry—are required [9,18,19,20,21,22,23,24,25].

Low-energy implantation has previously been demonstrated in 2D materials; however, these experiments were performed using broad beams with no control over spatial location [26,27]. Focused ion beams (FIB) have demonstrated the ability to target nanoscale features, enabling high spatial resolution, which has facilitated novel QIS experiments [28,29,30] but at relatively high energies (typically 35–100 keV). Additionally, using liquid metal alloy ion sources, a variety of ion species become available with isotopic selectivity, enabling QIS applications due to precise control over a defect’s hyperfine coupling [31]. Here, we demonstrate the combination of low-energy implantation with high spatial resolution using a newly developed biased sample plate technique combined with existing FIB capabilities.

While experiments have shown successful implantation into single atomic layers, there are few accessible simulation codes for ultra-low-energy implants [26,32,33]. The de-facto standard for simulating ion–solid interactions remains the “Stopping and Range of Ions in Matter” (SRIM), a binary collision approximation-based Monte-Carlo code [34]. The accuracy of SRIM calculations, however, has repeatedly been questioned, especially with respect to the electronic stopping model [35,36,37] and sputtering simulations [38,39]. Moreover, SRIM considers impact parameters up to π^−1/2^N^−1/3^ only, where N denotes the atomic density of the target. For sputtering simulations, “simultaneous weak collisions” have been introduced [40], which add collisions with larger impact parameters to the simulation. Recently, it has been shown that simultaneous weak collisions are also relevant to heavy ion implantation [41]. Lastly, SRIM does not consider dynamic changes to the lattice during implantation. Therefore, we use two additional binary collision codes—TRIDYN [42] and IMSIL [43,44,45]—to assess the accuracy of SRIM.

Here, we address implanting novel ion species at low energies and with precise positioning by demonstrating 1 keV Au implantation into Si using a Raith VELION FIB. The depth profile of the Au is then measured by a LEAP4000 XHR Atom Probe Tomography (APT) system. Our results show that the implant depth at 1 keV is 0.8 nm, and that low-energy ion implants can be achieved via lowering the column voltage and decelerating the ions using a biased plate. We estimate our current achievable spot size to be sub-micron, supported by SIMION simulations. We contrast experimental and static SRIM results with the dynamic binary collision codes TRIDYN and IMSIL, arriving at a more accurate model for the ion range but overestimating the ion straggle.

## 2. Methods

Au implantation was performed using a 35 kV Raith Velion FIB with lithography capability for nanoscale direct-write patterning. The typical targeting resolution was <35 nm with a beam spot size < 10 nm at 35 kV. We reduced the beam energy or landing energy (E) in two ways; (1) by reducing the source voltage (V_S_); and (2) by modifying the standard sample holder to allow biasing of the sample. The adapted sample holder consisted of the original holder, a biased plate held at a potential (V_P_), and a grounded plate with a 12 mm diameter hole. The landing energy E is calculated as E = q × (V_S_ − V_P_), where q is the charge state of the ion. The plates were fixed by stainless steel screws and insulated with Polyetheretherketone (PEEK) spacers. This technique allows us access to much lower energies than available to our FIB, which can only operate at landing energies as low as 1 keV. Using the bias plate approach, we can theoretically lower the landing energy to eV range.

The low-energy ion implants were carried out on a treated Sb-doped Si microtip array purchased from CAMECA Instruments. Our ion energies and fluences were chosen to elucidate the effects of the ion charge state and ion landing energy, as well as the effects of ion fluence on ion range. The tip preimplantation treatment consisted of running each Si tip in a LEAP4000 XHR APT in voltage mode at 50 K with a 20% pulse fraction and a 5 kV stopping voltage (described in more detail below). The preimplantation treatment was performed to increase the tip surface area and allow for more atoms to be collected from the shallow implant volume. After implantation, the APT experiments were run using the same conditions. The APT results were reconstructed and analyzed using CAMECA’s interactive visualization and analysis software (IVAS 3.8). The 400 Si pole lattice spacing was used to calibrate the voltage, tip evolution, and reconstruction algorithm to ensure an accurate depth profile.

The finite element method was used to simulate the temperature and fluid distribution. Combined with the simulation results, the force of the submicron particles in the vortex was calculated and analyzed to explain the trapping mechanism via vortex flow. The temperature and velocity distributions on the liquid surface were obtained through numerical simulation, and then the distributions along the long and short axis for one of the vortices were taken as examples. The velocity gradient and temperature gradient along the corresponding direction of long and short axes were calculated, and the fluid pressure (*F*_p_) caused by the velocity gradient and the thermophoretic force (*F*_T_) caused by the temperature gradient, as well as their net force (*F*_net_), was systematically obtained. The obtained results show that *F*_p_ tends to move the particle to the position of the local velocity maximum, while *F*_T_ tends to keep the particle away from the tip end. In the region near the tip end, *F*_T_ is much larger than *F*_p_, and the particles are pushed away from the tip end. In the region far away from the tip end, *F*_T_ is much smaller than *F*_p_, and *F*_p_ plays a dominant role in net force *F*_net_; thus, the particles are pushed to the position of the local velocity maximum. From the tip end to the vortex center, *F*_T_ is reduced to that comparable with *F*_p_ after a certain distance, and the combined action of *F*_T_ and *F*_p_ enables the particles to be trapped into a certain region between tip end and vortex center. Therefore, an elliptical annular region for particle trapping can be created, which successfully explains the experimental observation.

## 3. Results

Figure 1a illustrates the results of the low-energy Au implants into the Si APT pillars (see Section 2) using a Raith VELION FIB-SEM. The Au fluence at V_S_ = 1 kV (blue) was 10^16^ ions/cm^2^, and at V_S_ = 5 kV with V_P_ = 4 kV (red), 10^15^ ions/cm^2^. The solid cyan line represents SRIM calculation for 1 keV Au into Si. At V_S_ = 5 kV (red) and 10^15^ ions/cm^2^ fluence, we used a custom designed sample holder (Figure 1b) to decelerate ions to the desired landing energy. We tune the landing ion energy by adjusting the voltage on the biased sample plate using the high-voltage power supply.

The peak position for 1 keV landing energies was obtained using a Gaussian function fit of the APT datasets, shown as the dashed blue and the dashed red curves in Figure 1a, for implant condition with and without the biased plate, respectively. The depth extracted from the two fits was measured to be 0.86 ± 0.01 nm and 0.71 ± 0.09 nm. Both values closely agree, and the measurement error is smaller than a single Si lattice constant, indicating that the biased plate is effective at lowering the landing energy. An experimental comparison with SRIM was carried out by fitting the SRIM results with a Gaussian function. SRIM predicts the penetration depth of the Au beam into Si at 1 keV to be 4.1 nm, exceeding the experimental values by 4.7×, as was expected based on the low-energy and high-fluence conditions. As will be discussed at some length below, we are implanting ions in a very low-energy regime, where the static SRIM simulations are insufficient to accurately capture ion stopping due to dynamic alterations to the lattice.

To quantify the depth of the experimental implants, we used APT. Compared to other depth profiling techniques such as Rutherford backscattering spectroscopy or secondary ion mass spectrometry, APT has superior depth resolution limited only by atomic lattice spacing for Si in the Z-dimension. However, a drawback of APT is sample preparation, necessitating sharp conductive tips to enable field emission at the tip apex. We implanted Au into a commercially available APT tip coupon consisting of a 6 × 6 tip array. As-received tips have been found to be too sharp for the shallow implants to be detected. This is due to the small number of atoms 1–2 nm deep with the nanoscale surface area in the as-received Si microtips; specifically, the implanted volume of an as-received tip is only a few unit cells, and the number of implanted Au atoms collected from that volume is insufficient to produce proper mass spectra with statistically significant counts. To circumvent this issue, we pre-dulled the tips before Au implantation using the APT setup. The pre-dulling consisted of running the APT in voltage-mode with a 5 kV voltage stop to increase the tip surface area and create an almost-perfect spherical sector end form. SEM pictures of an as-received and pre-dulled tip are shown in Figure 2a,b, respectively, with the original tip shape denoted by white lines in Figure 2b. The almost-perfect spherical sector APT tip end also improves the accuracy of the reconstruction. After the APT measurement, a three-dimensional reconstruction of the atomic arrangement of the implanted tip was obtained, shown in Figure 2c, with Au atoms denoted by yellow spheres and Si atoms shown as black dots.

As a pathway towards ultra-low-energy (10–100 eV) FIB implantation, we assess the ion beam spot size in the biased sample holder using a simplified SIMION simulation [31,46] using a three-element Einzel lens with a 60-μm beam diameter matching the beam-defining aperture used in these experiments. The initial beam is assumed to be parallel and with a Gaussian FWHM energy spread of 15 eV, based on the typically reported values in the literature for AuSi eutectic-based liquid metal ion sources [47]. A focus solution for the 5 keV landing energy Au^+^ beam was found by varying the Einzel lens voltage in the SIMION program. The optimal beam spot was found to be 60 nm in diameter, as shown in Figure 3a, matching within a factor of two the actual beam diameter in the Raith Velion experiments. For demonstration purposes, in Figure 3, we only plot 12 of the 60 trajectories used in the simulation and increase the diameter of the incident beam to 600 μm to make them visible. Using the focus solution found for V_P_ = 0, V_P_ was then set to 4 kV, i.e., 1 keV ion landing energy, in the SIMION program, and the beam spot increased to 500 nm. Since experimentally refocusing the beam and measuring the spot size on the biased sample holder may not be feasible due to low secondary electron yield and reattraction of secondary electrons by the biased plate, we believe this to be the best spot size estimate of a well-focused beam on the 4 kV biased target. The result of the simulation is shown in Figure 3b. In future simulations, we will explore the possibility of further improving the spot size by using the top plate as a focusing element by applying a small bias to that plate.

## 4. Discussion

To address the large discrepancy between the experimental implant depth and the depth predicted by SRIM simulations, we employed two additional binary collision codes: TRIDYN and IMSIL. Furthermore, we measured the same range for a fluence of 10^15^ ions/cm^2^ and 10^16^ ions/cm^2^ and a change of Au enrichment by only 2×. Based on the IMSIL and TRIDYN simulated data showing the ion distribution shifting to the surface with the increase in fluence, we suspect that turning on the HV plate led to an erroneous estimation of the ion fluence, and that our 10^15^ ions/cm^2^ fluence is close to a fluence of 10^16^ ions/cm^2^ estimated from simulated ion ranges. To help identify the two different datasets, we continue to label them as 10^15^ and 10^16^ ions/cm^2^. The results from all implant conditions and simulations are summarized in Table 1.

TRIDYN allows us to perform ion irradiation simulations accounting for compositional changes [48]. Here, we modeled the collision cascade of Au ions with a given kinetic energy implanted into a 300 nm Si slab. Nuclear stopping originates from the energy transfer to target atoms in consecutive binary collisions. The electronic stopping was equipartitioned between non-local (Lindhard-Scharff [49]) and local (Oen-Robinson [50]) contributions. In the static version (i.e., without taking compositional changes into account), the depth distribution of the implanted Au ions is extracted from a set of 100,000 ion impacts and shown as the cyan curve in Figure 4. Contrary to the experimental results with much shallower distributions, static simulations (reproducing SRIM) predict implantation ranges of about 5 nm for 1 keV Au ions. These static simulations do not account for surface relaxation and compositional changes in the target material. Switching to dynamic calculation, which allows us to model compositional changes and implements beam-mediated diffusion, provides qualitative agreement with the experiment for larger ion fluences. However, the simulation of 10^15^ ions/cm^2^ does not exhibit significant difference when compared to the static calculation; even a calculation at 10^16^ ions/cm^2^ does not reproduce the experimentally observed ion range. Only when increasing the fluence to 3 × 10^16^ ions/cm^2^ is qualitative agreement of the implant shape found. Note that temperature-mediated diffusion is not included in this approach, and temperature effects should cause the distribution to broaden and move towards the surface. The ion straggle is already overestimated when compared to SRIM and the experiment, as shown in Table 1. Including an additional broadening mechanism from thermal diffusion will further increase the estimated straggle.

Additionally, we used the “Implant and Sputter Simulator” (IMSIL) code to model surface sputtering and compositional changes to the lattice. In IMSIL, by default, the universal ZBL interatomic potential and electronic stopping with an equipartition rule (as in TRIDYN) is used, with a correction factor of 1.2 for Au in Si. Results comparing IMSIL simulations to experimental data are shown in Figure 5a. Like TRIDYN, the shape of the measured implant profile is not fully captured by the simulation, but the overall range is accurately predicted for a simulated fluence of 10^16^ ions/cm^2^. Figure 5b explores varying IMSIL parameters to match SRIM simulations more closely in the static limit. IMSIL simulations, by default, use energy-dependent maximum impact parameters [45], p_max_, which evaluate to 1.95 Å at 1 keV and 3.15 Å below 60 eV for a Au ion slowing down in Si (this replaces the “simultaneous weak collisions” model of other codes [40] with a continuous energy-dependent model). It is also possible to use a fixed p_max_, which leads to results much closer to SRIM, if the SRIM maximum impact parameter of 1.53 Å is used (blue vs. cyan curve). On the other hand, changing p_max_ to larger values did not change the IMSIL results further. To explain the remaining difference between IMSIL and SRIM, we searched for other parameters which might be different between the two codes. Two other default models of IMSIL, which are different from SRIM, are the use of the time integral in the calculation of the turning points of the ion trajectories [51] and of statistical free flight paths distributed according to a log function. Switching these two options off moves the simulated profile deeper into the bulk (effect of the time integral) and sharpens the profile (as a result of the statistical free flight path).

The influence of the maximum impact parameter indicates that the interaction of the ion with more distant target atoms leads to the additional slowing down of the ion. It also indicates that an accurate model of the interatomic potential at large separations is important. In some codes, the so-called Kr-C potential [51] is used, while SRIM and IMSIL use the universal ZBL potential by default. Using the Kr-C potential in IMSIL (red curve in Figure 5b) instead of the ZBL potential (green curve) shifts the profile to larger depths. While we cannot decide within this study which interatomic potential is more accurate, we may conclude that the difference is quite significant.

Lastly, we compare the IMSIL and TRIDYN simulations at 10^16^ ions/cm^2^ to our experimental values. We find that IMSIL and TRIDYN are in excellent agreement; however, both codes predict a long tail with a half-width at half maximum close to the SRIM prediction. This indicates that even though significant sputtering is happening in the sample, the maximum Au range is still close to that predicted by SRIM, which is contradicted by our data having an HWHM of approximately 0.7 nm.

While experiments in the non-sputtering regime would make for an easier comparison between various simulation codes, the small area of the APT tip requires a high implantation fluence to obtain sufficient Au atom statistics at various depths. Although APT relies on field emission and time-of-flight mass spectrometry, only 33% of ions are collected. Additionally, when clusters of atoms are removed, the measured depth is typically shallower than the real depth, a potential explanation for our data having significantly shallower straggle than predicted by both TRIDYN and IMSIL. Further experiments will need either to take into account the effects of ion sputtering or use alternate means of ion detection; for example, through the use of scanning tunneling electron microscopy for detecting individual impurity atoms.

## 5. Conclusions

We have implanted Au ions at landing energies of 1 keV and verified the implant depth using APT. Our results show that the implant depth at 1 keV is 0.8 nm, in stark contrast to SRIM simulations, which predict implant depth of 4.1 nm at 1 keV. This discrepancy between experimental and SRIM-modeled implant depth is largely resolved through modeling with dynamic binary collision approximation-based codes: TRIDYN and IMSIL. While some uncertainties in the binary collision model remain, these codes show that the main mechanisms behind the shallow, experimentally measured implant profiles are the enrichment of the Si lattice with higher-stopping-power Au and surface sputtering, leading to an apparently shallower implant. A focus solution using the design of the decelerated VELION system was found using SIMION. For 5 keV Au^+^, a beam spot of 60 nm was calculated, and using the same bias on the objective lens of the VELION, increasing the bias on the target to 4 kV, the spot size increased to 500 nm. Beyond the data shown in the manuscript, we also performed implantation at 2 keV landing energy and saw a similar discrepancy between SRIM simulation and experiment. We are able to qualitatively reproduce the results through dynamic simulations using TRIDYN and IMSIL, similar to the results at 1 keV.

## Figures and Tables

**Figure 1 micromachines-14-01884-f001:**
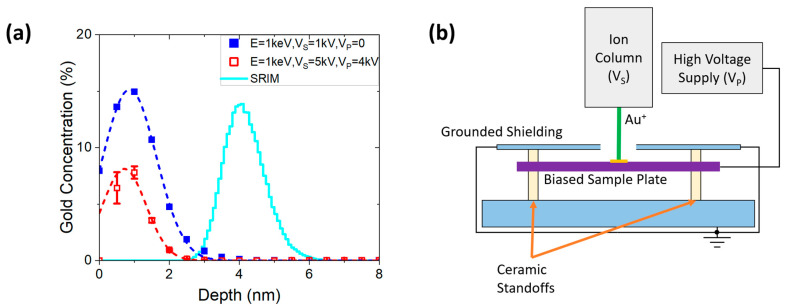
Low-energy implant results with and without biased sample holder. (**a**) Implant depth profile measured via APT (with data points fitted with a Gaussian) (dashed line) and calculated via SRIM (solid line). The blue squares denote an implant that maintains the sample at ground, while the red open squares denote the use of the biased sample plate to decelerate ions to the desired landing energy. Results of V_S_ = 1 and 5 kV are shown in blue and red, respectively. (**b**) Schematic of the custom designed sample holder used for decelerating ions.

**Figure 2 micromachines-14-01884-f002:**
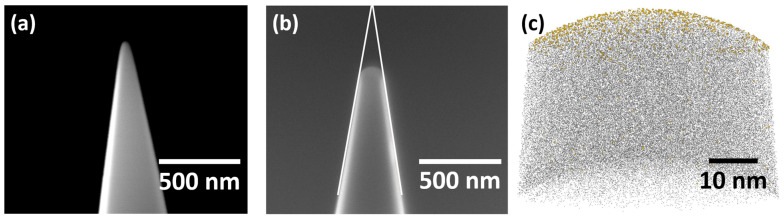
APT sample preparation and results. (**a**) SEM image of an APT tip prior to dulling. (**b**) Tip after being dulled in the APT setup ready for ion implantation. The sharp tip must be pre-dulled in the APT setup to maintain the shape of a spherical sector. The original structure of the tip is shown by white lines. (**c**) Reconstructed tip composition. Si ions are shown in black and Au ions are shown in yellow.

**Figure 3 micromachines-14-01884-f003:**
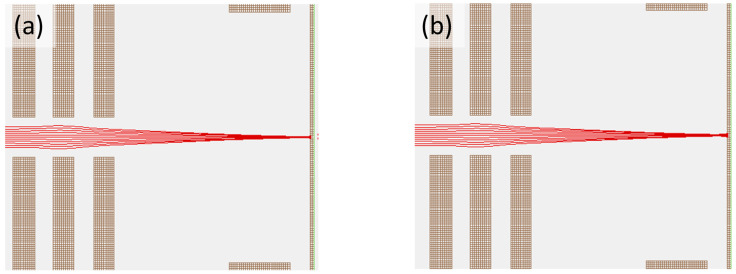
Representative SIMION calculations to determine the ion beam spot size. (**a**) SIMION calculation for a 5 keV 60-μm-diameter parallel Au^+^ beam focused to a 60 nm spot. (**b**) Calculation for a 5 keV Au^+^ beam with V_P_ = 4 kV applied to the sample plate with the remaining ion optics unchanged. The spot size increases to 500 nm. The images in (**a**,**b**) contain only 12 ion trajectories for illustration purposes and are for a beam that has a 600 μm diameter.

**Figure 4 micromachines-14-01884-f004:**
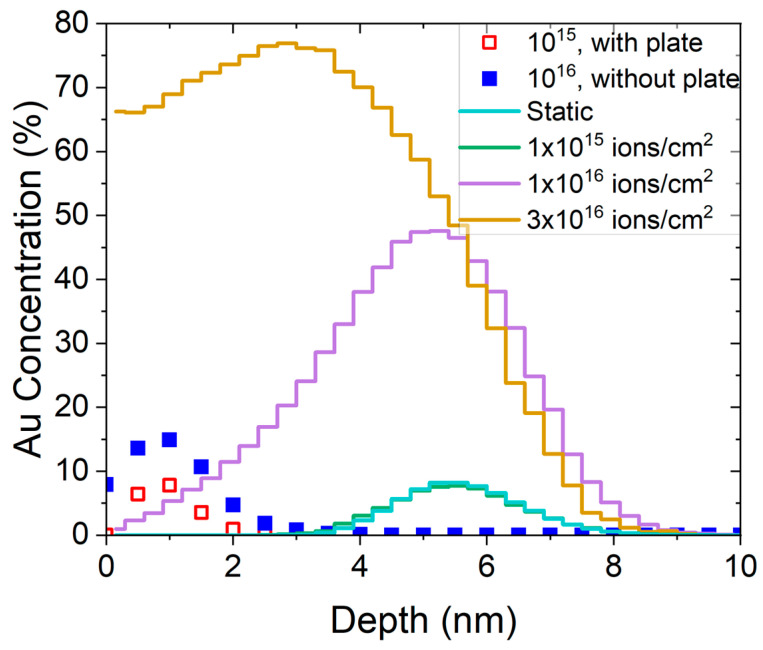
Implantation profiles of Au-irradiated Si obtained via BCA simulations with the TRIDYN program (solid lines) compared to experimental data (rectangles). The static calculation reproduces the SRIM simulation, assuming an implantation fluence of 10^15^ ions/cm^2^.

**Figure 5 micromachines-14-01884-f005:**
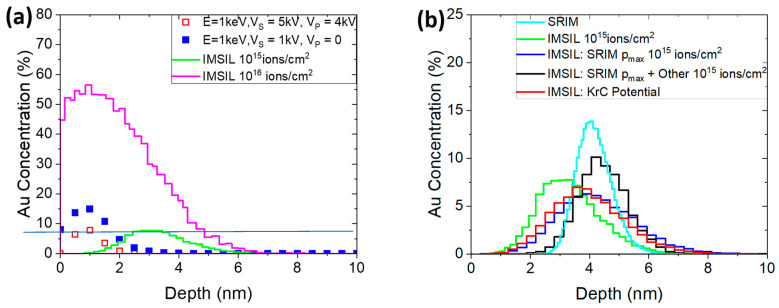
(**a**) IMSIL simulations reproducing experimental data. (**b**) IMSIL simulations reproducing SRIM predictions by varying the impact parameter and excluding diffusion from the simulation.

**Table 1 micromachines-14-01884-t001:** Experimental implant depth and simulated range comparisons for low-energy Au ions into Si. The straggle is measured as the full-width half maximum from a Gaussian fit.

V_S_	V_P_	Energy	Fluence	Measured		SRIM		TRIDYN		IMSIL	
				Depth	Straggle	Depth	Straggle	Depth	Straggle	Depth	Straggle
kV	kV	keV	ions/cm^2^	nm							
1	0	1	10^16^	0.86	0.76	4.10	0.58	4.95	1.60	1.51	1.58
5	4	1	10^15^	0.71	0.62			5.49	1.03	3.20	1.00

## Data Availability

The data that support the findings of this study are available from the corresponding author upon reasonable request.

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
