# Peer review of "Measurement and Simulation of Ultra-Low-Energy Ion–Solid Interaction Dynamics"

_micromachines, 2023, doi:10.3390/mi14101884_

Round 1
Reviewer 1 Report
In this manuscript, the authors introduce a method to implant Au ions to depths less than 1 nm in silicon while utilizing APT simulations and binary collision approximation codes to verify these findings. These findings also illustrate the importance of considering dynamic processes while simulating the stopping range of ions in matter since static simulations are not sufficient to model these interactions. These findings are of particular importance for emerging novel materials and devices as well as the continuing miniaturization of semiconductor technology.
The authors discuss their findings in a clear and concise manner with a thorough discussion highlighting their findings and explaining discrepancies between implant depth calculations with SRIM and their experimental findings. I recommend that manuscript be published in Micromachines following some minor revisions.
1. In line 178, I believe there is a typo where “second electron” is stated in place of “secondary electron”.
2. Is crystallographic orientation considered in the simulations? Since the crystallographic orientation has numerous effects including the channeling profile of the ion species this should be clarified in the discussion.
3. I believe it would be beneficial to briefly discuss the impact of the ion species on the implantation depth. Can this approach be used for shallow implantation of elements lighter than Au?
Reviewer 2 Report
In the work of Titze et al. a FIB system with a Au source is used to implant 1 KeV ions into Si. The experiments were supported by some Montecarlo simulations from three different models from 1984. The work is suitable for the level of the journal
Reference 33 doesn't relate to IMSIL as it does 100% experimental lifetime experiments but I guess it helps with the self citation farming.
44% self citations: 3 Hobler + 3 Bielejec + 10 Belianinov + 2 Krasheninnikov.
More interesting concerns:
1) What’s the range of low energy implantation? There are papers implanting 20 eV W ions (https://doi.org/10.1038/s41699-022-00318-4).
2) Does the pre-milling affect the implantation profile as the tip could become amorphous?
3) Silicon on gold from a 50 nm tip is a good HAADF contrast for HRSTEM, (https://doi.org/10.1016/j.ultramic.2015.02.011) authors are encourage to showcase with microscopy the implantation of the ions to improve the quality of the paper instead of promising future experiments for more citation farming.
4) The mismatch between experiments and simulations is irreconcilable, even modifying parameters to match the observed profile results on a wrong concentration level. Moreover, it is difficult to be sure with just one experimental dataset, if this were another journal it wouldn’t pass, but as the authors are honest with their limitations it is acceptable here.
Check the format of your references, when a doi is added is duplicating the info adding the web browser complete address.
Typo on last line of 2 paragraph of methods.
On section 3 it points out that Details of the sample holder design are discussed in the Methods section (section 2), so redundant info of what is already explained.
"a SEM picture of an as-received tip is shown in Figure 2a" This line is rarely lying there and disturbing the readability of the text, should be worded differently to improve the flow of the story/description.
There are other small issues with the text, with so many authors it should be written better.
Author Response
Please see the attachement.

Round 2
Reviewer 2 Report
The decision to accept publication is left to the editor.
-